# Synthesis and Biomedical Applications of Highly Porous Metal–Organic Frameworks

**DOI:** 10.3390/molecules27196585

**Published:** 2022-10-05

**Authors:** Ahmed Ahmed, Darragh McHugh, Constantina Papatriantafyllopoulou

**Affiliations:** SSPC the Science Foundation Ireland Research Centre for Pharmaceuticals, School of Biological and Chemical Sciences, College of Science and Engineering, University of Galway, University Road, H91 TK33 Galway, Ireland

**Keywords:** metal–organic frameworks, biomedical applications, highly porous, biomolecules, drug delivery

## Abstract

In this review, aspects of the synthesis, framework topologies, and biomedical applications of highly porous metal–organic frameworks are discussed. The term “highly porous metal–organic frameworks” (HPMOFs) is used to denote MOFs with a surface area larger than 4000 m^2^ g^−1^. Such compounds are suitable for the encapsulation of a variety of large guest molecules, ranging from organic dyes to drugs and proteins, and hence they can address major contemporary challenges in the environmental and biomedical field. Numerous synthetic approaches towards HPMOFs have been developed and discussed herein. Attempts are made to categorise the most successful synthetic strategies; however, these are often not independent from each other, and a combination of different parameters is required to be thoroughly considered for the synthesis of stable HPMOFs. The majority of the HPMOFs in this review are of special interest not only because of their high porosity and fascinating structures, but also due to their capability to encapsulate and deliver drugs, proteins, enzymes, genes, or cells; hence, they are excellent candidates in biomedical applications that involve drug delivery, enzyme immobilisation, gene targeting, etc. The encapsulation strategies are described, and the MOFs are categorised according to the type of biomolecule they are able to encapsulate. The research field of HPMOFs has witnessed tremendous development recently. Their intriguing features and potential applications attract researchers’ interest and promise an auspicious future for this class of highly porous materials.

## 1. Introduction

Metal–organic frameworks (MOFs) are a family of hybrid porous materials that have attracted considerable research attention in recent years due to being promising candidates in a variety of significant applications, including sensing, catalysis, drug delivery, spintronics, photonics, and others [1,2,3,4]. MOFs display a plethora of aesthetically pleasing framework topologies, as well as unique structural features, such as large surface areas, tuneable porosity [5,6], and the possibility of targeted introduction of functional groups into their framework [5,6,7,8,9]. In regard to their porosity, MOFs are classified as microporous, mesoporous and macroporous according to their internal pore width. This categorisation is in line with the IUPAC classification of porous materials, which correlates the confined pore size to the encapsulated guest molecules during a physisorption process [10]. According to this definition, the vast majority of MOFs are microporous (pore width < 2 nm), a few of them are mesoporous (pore width between 2 and 50 nm), while examples of MOF composites in the macroporous range (pore width >50 nm) have been reported recently [11,12,13].

MOFs can encapsulate various guest molecules, ranging from small cations [14], to large organic molecules, e.g., drugs [15], dyes [16], pharmaceutical and personal care products [17,18,19], etc., which makes them suitable to address contemporary challenges in the biomedical, environmental, and industrial field. In particular for applications that involve the encapsulation of biomolecules, or other bulky guest molecules, e.g., enzyme immobilisation [20,21], protein crystallisation [22], and gene targeting [23], the isolation of highly porous MOFs (HPMOFs) is essential. The latter refers to MOFs with large pores, and, in this review, these are specified to MOFs that possess surface area higher than 4000 m^2^ g^−1^. HPMOFs are advantageous in comparison to other porous materials (liposomes, zeolites, quantum dots, etc.) especially when it comes to the encapsulation of large guest molecules [24]. For example, zeolites are microporous with their pore diameter being typically less than 13 Å, making these materials inadequate in the encapsulation of large molecules [25]. Liposomes, although biocompatible and able to encapsulate a large amount of guest molecules, they have poor in vivo stability and display rapid removal from the bloodstream by the phagocytic cells of the reticulo-endothelial system [26]. On the contrary, porous organic polymers (POPs) have tuneable biological properties and hierarchical porous structures, and they can be pH-responsive, which deem them promising drug delivery systems [27,28]. POPs display similar attributes to HPMOFs in regard to their high surface area, porosity and functionalisation; yet, they can contain weak linkages and tend to hydrolyse in physiological conditions, which is a setback especially for their use in biomedical applications [29].

There is a linear correlation between porosity and surface area in HPMOFs [10]; however, although this rule is widely applicable, there can be exceptions as the porosity is affected by the degree of framework interpenetration that takes place. This happens when more than one networks are intertwined or entangled in each other resulting in the decrease of the pores’ size [30,31]. The interpenetration effect, that cannot be easily controlled synthetically [30,32], along with their poor stability [33], are two of the main challenges toward the development of new HPMOFs.

More than 100,000 MOFs are now available in the Cambridge Structural Database [33]. The isolation of such species has led to the development of numerous efficient synthetic approaches, including solvothermal techniques [34], post-synthetic [35], isoreticular [36], microwave-assisted [37], surfactant assisted [38], mechanochemical [39] and sonochemical synthesis [40], multivariate [41], mixed-ligand [42], pillared approaches [43], and others; yet the development of efficient synthetic approaches towards HPMOFs remains of ultimate importance. A lot of research has been taking place recently in order to gain synthetic control in the interpenetration degree and stability of HPMOFs [44,45], which has led to the development of new synthetic strategies, including the use of metal–organic polyhedra (MOPs) as building blocks [46,47], controlled defect formation [48,49], and the employment of space-efficient moieties. For instance, it has been demonstrated that the replacement of bulky moieties in the framework by space-efficient ones has a profound impact on the surface area. Thus, the replacement of phenyl groups by acetylene units resulted in the increase of the computationally determined maximum limit of surface area from ~10,000 m^2^ g^−1^ to ~14,600 m^2^ g^−1^ [7].

The review contains four sections: the first one is introductory, the second attempts to categorise some of the main synthetic approaches towards HPMOFs, the third describes HPMOFs with capability to encapsulate large biomolecules, along with description of the encapsulation strategies, and the last one contains summary and concluding remarks for this area of chemistry. More than 40 HPMOFs have been reported and discussed herein; these are listed in Table 1 along with brief information about their surface area, porosity and design approach. This paper aims to include a discussion for every HPMOF, emphasising on synthetic and framework topology aspects, and to provide the reader with some idea of the range of biomedical applications of HPMOFs and the chemistry that has been carried out in this area.

## 2. Synthesis

The main synthetic challenge in the synthesis of HPMOFs is their low stability. In order to combat this issue, polytopic rigid linkers [84] (Figure 1, Figure 2 and Figure 3) that favour strong coordination bonds are employed for their construction [84,85,86]. Thus, aromatic phenylene or acetylene-based linkers are traditionally used in combination with strongly coordinating carboxylate linkers and high oxidation state metal ions [52]. Coordination between carboxylates and high oxidation state metals results in the most thermodynamically stable M-L bonds, thus providing enhanced stability to HPMOFs [87]. Numerous methods have been employed in the synthesis of HPMOFs and are discussed below, including isoreticular expansion [88,89,90], which is the most common method, as well as topological control [91], use of metal–organic polyhedra (MOPs) as building blocks [54], and controlled defect formation [92,93].

### 2.1. Isoreticular Expansion

Isoreticular expansion involves the extension of the length of a linker whilst retaining its symmetry and geometry [88,89,90]. This leads to a MOF of the same topology but with larger pore dimensions than the parent MOF. The extension of the linker requires a rigid group, usually acetylene or phenylene, in order to maintain the same topology. This has led to notable families of MOFs with progressively increasing pore dimensions. The first example of isoreticular expansion was the IR-MOF family, where substitution of the initial BDC (1,4-benzenedicarboxylic acid), for BPDC (biphenyl-4,4′-dicarboxylic acid) (IR-MOF-10) and TPDC (IRMOF-16), with pore diameter expanding from 18.5 Å for IRMOF-1, 24.5 Å for IRMOF-10 and 28.8 Å for IRMOF-16 (Figure 1) [63,88,94]. This has similarly been carried out on UiO-66 to synthesise MOFs UiO-67,68, with largest cage diameters of 11 Å, 16 Å and 25.6 Å, respectively [95]. Isoreticular chemistry has been also extended to linkers with higher topicity, including tritopic (HKUST-1 analogues [88,96,97,98], tetratopic (NU-1100-NU-1104) [70,99,100,101] and hexatopic linkers (PCN-61 analogues [66,102,103], NU-1500 family [73,74,104]) (Table 1).

Reticular chemistry is a proven method of synthesising highly porous MOFs, however numerous issues arise once sufficient linker length is reached. These include poor crystallinity [105], the collapse of framework upon desolvation [106], and interpenetration. [107,108,109,110] The likelihood of interpenetration increases as linker length increases, with increasing interpenetration typically resulting in reduced pore volume and even non-porous materials [111]. A representative example of this the IR-MOF family where IR-MOF-9 is a doubly interpenetrated version of IR-MOF-10, IR-MOF-9 has a BET surface area of 900 m^2^ g^−1^, lower than that of IR-MOF-10 (1600 m^2^ g^−1^) [111,112]. Interpenetration can be desirable for specific applications such as gas storage, where increased interpenetration can increase the number of binding sites for a gas per unit volume [113,114,115,116]. Numerous synthetic methods have been developed to prevent or limit interpenetration, including use of bulky ligands [117], covalent attachment of bulky groups to the ligand which can then be removed post-synthetically [32,118,119], additives [115] and reaction conditions with more dilute solutions yielding less interpenetrated frameworks [88,120].

Whilst increasing linker length via isoreticular expansion has been thoroughly studied as a means of synthesising HPMOFs, the use of larger SBUs has received less attention [121,122,123,124]. Larger SBUs result in MOFs with higher connectivity, and thus larger pore size. An increase in SBU size also decreases the likelihood of interpenetration. An excellent example of this methodology is bio-MOF-100 (Zn_8_(ad)_4_(bpdc)_6_O_2_•4Me_2_NH_2_, 49DMF, 31H_2_O) (ad = adeninate), where the combination of a known zinc-adeninate octahedral building unit (ZABU) with a bpdc^2−^ linker resulted in a framework with a surface area of 4300 m^2^ g^−1^. ZABU consists of 8 Zn^2+^ ions connected through four adeninates and two oxo groups (Figure 2).

A size comparison of the ZABU to the archetypical [Zn_4_O(COO)_6_] SBU shows a significant increase in pores size from 10.5 Å to 14.2 Å. The connectivity is also significantly increased with each ZABU being 12-connected in contrast to the 6-connected [Zn_4_O(COO)_6_] SBU. Each ZABU is connected to four other units via two BPDC^2−^ ligands, generating a framework with large 28 Å diameter cylindrical channels. It is worth mentioning that isoreticular chemistry has been applied to bio-MOF-100 resulting in significantly increased pore sizes. Bio-MOF-101 and bio-MOF-102 were prepared by stepwise linker exchange from bio-MOF-100, with an increase in surface area from 2701 m^2^ g^−1^ to 4410 m^2^ g^−1^ [51].

The use of polyoxometalates as SBUs in MOFs is of interest due to the retention of their catalytic [125,126], medicinal [127], photochemical [128] and electrochemical properties [129] and has produced similar results as above, with MOF-688 being an excellent example [128,130,131,132,133,134]. MOF-688 is an imine linked MOF derived from the amine-functionalised polyoxometalate, [N-(C_4_H_9_)_4_]_3_[MnMo_6_O_18_{(OCH_2_)_3_{CNH_2_}_2_] and the tetrahedral linker tetrakis(4-formylphenyl)methane (tpim). The framework adopts a 3-fold interpenetrated dia topology, where the organic linkers are the vertices of the framework and the polyoxometalate acts as ditopic linker. Whilst the pore volume of MOF-688 is substantially reduced, this could be avoided by the synthesis of a non-interpenetrated framework [135].

### 2.2. Topological Control

Topological control involves the a priori combination of SBUs and linkers of desired connectivity and geometry to obtain MOFs of precise topology with a low degree of interpenetration. The selection of linker and its binding mode, metal oxidation state, and reaction conditions play a crucial role in the topological control synthetic approach [136,137,138,139]. This method can be combined with isoreticular approach to generate MOFs of higher porosity; indeed, the MOFs with the largest surface areas are obtained via a combination of reticular chemistry and topological control [7,60,64,69,70,71,75,77]. For instance, by selection of topologies that reduce or inhibit interpenetration, isoreticular MOFs can be synthesised with significant increase in pore volume [66,102,140,141]. This is the case in the IR-MOF-74-I to XI series, where the use of a dioxidoterephthalate (DOT) linker led to the formation of an etb net MOF with infinite rod [M_3_(O)_3_(-CO_2_)_3_]_∞_ SBU (M ^=^ Zn^2+^, Mg^2+^) [60,140,142]. The rods are in a hexagonal arrangement with tightly packed DOT^3-^ bridging them. This results in a hexagonal channel- type MOF with the isoreticular series all yielding non-interpenetrated structures (Figure 3). The pore apertures were increased from 14 to 98 Å for IR-MOF-74-I to IR-MOF-74-XI.

Higher connectivity linkers can also decrease the likelihood of interpenetration, allowing for the synthesis of HPMOFs. A representative example of this is the isoreticular series PCN-61, PCN-66, PCN-68 and PCN-610, which are constructed from dendritic C_3_ symmetrical hexacarboxylate linkers and Cu_2_ paddlewheel SBUs forming a (3,24) connected net with BET surface areas up to 6000 m^2^ g^−1^ [103,142]. The use of coplanar isophthalate linkers results in a fixed distance between SBUs. This fixes the window size between tetrahedral and octahedral cavities of the framework, thus extension of linker length will result in expansion of the tetrahedral and octahedral cavities without interpenetration or collapse. This method has been extended to numerous MOFs displaying high surface areas and pore volume with the MOF NU-110E having the largest surface area of 7140 m^2^ g^−1^ to date [7].

Similar to the use of high connectivity linkers, high connectivity SBUs lead to MOFs with high porosity. For example, Zr carboxylate MOFs are some of most highly porous, featuring SBUs with high connectivity along with highly stability [143,144,145]. The isoreticular family NU-1101–1104 share an ftw topology with almost square planar tetratopic linkers bridging the 6-connected Zr oxocluster SBU [71]. The ftw topology inhibits interpenetration due to the large square planar core of the linker being in the same plane as the linker carboxylates [146,147,148]. This allows for extension of the ligand arms, which results in progressively larger pore sizes, from 17.2 to 24.2 Å for NU-1101 and NU-1104, respectively (Figure 4). 

Certain topologies are more prone to the formation of highly interpenetrated nets. Specifically, MOF topologies with low linker and SBU connectivity such as pcu and dia are more prone to interpenetration than those with high connectivity [149,150,151,152,153,154]. MOFs with srs topology with up to 54-fold interpenetrated nets have been reported [155].

### 2.3. Use of Metal–Organic Polyhedra (MOPs) as SBUs

The use of metal–organic polyhedra (MOPs) as supermolecular building blocks (SBBs) has attracted great attention in the synthesis of HPMOFs [54,66,156]. MOPs provide many advantages over the traditional method of MOF synthesis; these include the control over the size and shape of the smallest pore, which is within the MOP, accessibility to pores via the linker connecting multiple MOPs, as well as control over the symmetry and size of the largest pore [46]. Many different geometries of MOPs have been reported as both discrete molecules and as part of extended solids such as MOFs [157,158,159]. The synthesis of MOPs requires ligands with specific topicity and binding geometry. In particular, linkers of tritopicity or greater are required to generate MOPs in extended solids, with the angle between adjacent coordination sites playing a vital role and must be less than 180° [46,79]. The adjacent coordination sites are used to generate the MOP. MOPs can then be connected via the opposite end of the linker or a secondary bridging linker.

One example of this approach is PCN-82 reported by Wu et al. PCN-82 is based on a dinuclear copper paddle wheel SBU and tetratopic [9,9′-(2,5-dimethoxy-1,4-phenylene)bis-(9H-carbazole-3,6-dicarboxylic acid)] (H_4_dpbcd) linker [79]. The linker features carboxylate donors, which are 90° from the adjacent carboxylate, allowing for the formation of the MOP. The linkers combine with the Cu paddlewheel SBUs to generate a 12-connected octahedral MOP (Figure 5). 

The MOPs are edge connected to each other via the phenylene unit of the linker. It is worth noting that as the MOPs are connected via phenylene units, the distance between MOPs can be extended, thus the pore size increases via reticular chemistry [160]. Three different cage sizes are present with dimensions of 11 Å, 12.9 Å and 18.1 Å, respectively. The MOP has also previously been isolated as a discrete molecule [161]. PCN-82 has a BET surface area of 4488 m^2^g^−1^, which is exceptionally high for a microporous MOF, establishing the usefulness of MOPs for achieving HPMOFs.

Another excellent example of how the MOP method allows for the formation of MOFs with substantially increased porosity and surface area without the use of extended linkers is [Cu_6_O(TZI)_3_(H_2_O)_9_(NO_3_)]_n_‚(H_2_O)_15_ (H_3_TZI = tetrazolylisophthalic acid) [162]. The framework is composed of a 24-connected SBB featuring twelve Cu_2_ paddlewheels bridged by isophthalate linkers and 3-connected Cu_3_(N_4_CR)_3_ tetrazolium based SBUs. The framework displays a BET surface area of 2847 m^2^ g^−1^ which is exceptional for a linker of this size. The framework is also interesting due to the presence of two different Cu SBUs, one to generate the SBB and another to bridge the SBBs. MOP-based MOFs have also been reported with linkers other than carboxylates. Fe-HAF-1 reported by Chiong et al. is a water stable MOF composed of Fe^3+^ hydroxamate polyhedral [163]. Specifically, Fe-HAF-1 is composed of tetrahedral Fe^3+^ hydroxamate MOPs bridged by the biphenyltetrahydroxymate (bpth^4−^) linker. The MOP has an internal diameter of 7 Å with a 16 Å diameter pore between MOPs. The framework shows excellent stability in organic solvents and a wide range of aqueous pHs. Quite rare among MOFs, Fe-HAF-1 is anionic at pH below 4 and displays selective uptake of cationic dyes. The large diversity of MOPs and their respective linkers is very promising for their incorporation into MOFs for a wide range of applications [164,165,166].

### 2.4. Controlled Defect Formation

Whilst all crystals inherently carry defects, control of defect formation in MOFs can be advantageous and allow for the synthesis of MOFs with larger, hierarchical porosity and pore size [92,93]. Defects in MOF crystals can be predesigned via modulational or engineered post-synthetically [167,168]. The use of mono-carboxylic acids as modulators has been used to create both missing linker and missing cluster defects in UiO-66 type MOFs [93,169,170]. The missing linkers and/or clusters leave void spaces in the framework and thus generate larger pores. The use of long chain alkyl monocarboxylic acid and insufficient concentrations of ligand yielded UiO-66 MOFs with mesopores and macropores [171]. MOF stability was retained despite the high defectivity. This method has also been shown to work on a wide range of MOFs with retention of chemical and thermal stability [172,173,174].

Whilst the pre-design of MOF defects is attractive due to the relative ease and versatility, post- synthetic methods allow for greater control of defect formation [175,176]. The introduction of pro-labile linkers and their subsequent removal allows for the introduction of larger and hierarchical pore structures [168]. For example, PCN-160 is composed of Zr_6_ clusters and azo-benzenedicarboxylate (azdc^2−^) (Figure 6). The azdc linker can be substituted by the prolabile imine linker 4-carboxybenzylidene-4-aminobenzoate (cbab^2−^) as the two linkers share a similar length and geometry [92,177]. The cbab^2−^ linker has been shown to be hydrolysed into 4-formylbenzoate and 4-aminobenzoate in the presence of acid. 

The linker exchange was achieved from 5.5% to 100%, however, only those where exchange between 0 and 43% were used for the linker labialisation and were treated by varying concentrations (0.5 M to 2 M) of AcOH as higher exchange % resulted in framework collapse. Following linker exchange and hydrolysis, the previously microporous MOFs show the presence of both micro and mesopores. The porosity was found to depend on both the % linker exchange and the concentration of AcOH used for linker hydrolysis. Increasing exchange ratio led to the formation of 25 Å mesopores between ratios of 0–9%, further increase results in the formation of 54 Å mesopores. This is accompanied by a decrease in the micropore volume, indicating the growth of micropores into mesopores. Increasing concentration of AcOH, resulted in a larger fraction of mesopores over micropores, this resulted in a decrease in surface area at the same linker exchange ratio. The largest pore (180 Å) was observed for exchange ratio of 43% and 2 M AcOH. The pore size could be varied between 10 Å to 180 Å by selection of linker exchange ratio and concentration of AcOH. The missing linker defects were also accompanied by missing SBU defects.

This method has similarly been applied to the SBU via the use of mixed-metal MOFs, where one metal is water sensitive. [M_3_F(bdc)_3_tpt] (tpt = 4,6-tris((4-pyridyl)triazine)) (M = Zn^2+^/Ni^2+^) is a microporous MOF where the Ni^2+^/Zn^2+^ ratio could be varied without change in the structure [178]. Exposure to water allows for the selective removal of Zn^2+^ and the coordinated ligands. The missing ligands and SBUs resulted in the formation of a hierarchical pore system with the appearance of 20 nm mesopores as the initial quantity of Zn^2+^ increases (Figure 7). 

Similar to PCN-160, the increase in mesopore volume coincides with a decrease in micropore volume indicating the growth of mesopores from the micropores. The hierarchical porous frameworks were compared to the original in the encapsulation of the cationic dye methylene blue (MB). The untreated MOFs showed negligible loadings, whilst those with the highest Zn^2+^ content showed increasing loading after treatment.

Other methods have been shown to allow for controlled defect formation, including thermolysis of the linker to form hierarchical pores along with the formation of ultra-small metal nanoparticles [179]. Selective photolysis of linkers to generate hierarchical porous MOFs has also been reported, where multivariate MOFs containing one photolysable linker and one photostable linker are synthesised. Subsequent exposure to a laser of appropriate wavelength decomposes the photolysable linker leaving a hierarchical porous framework [180]. These methods have allowed for what are initially microporous frameworks to encapsulate enzymes whilst still containing micropores for substrate diffusion [92,167].

**Figure 7 molecules-27-06585-f007:**
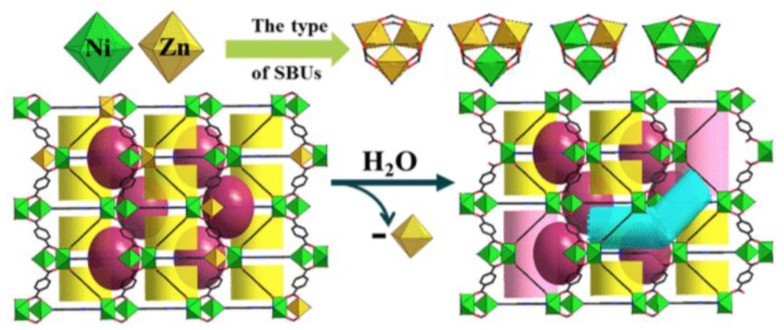
Generation of hierarchical porous framework via selective SBU hydrolysis in [M_3_F(bdc)_3_tpt] [178]. Adapted from ref. [180] with permission.

## 3. Large Biomolecules

The integration or encapsulation of large biomolecules into MOF pores has attracted significant attention as the resulting aggregates are promising candidates in antitumour treatment [181], gene delivery [182], biomolecular sensing [183], food processing [184], catalysis [185], and other applications [186]. HPMOFs have been used to encapsulate a range of biomolecules, including drugs, peptides [187,188], lipids, RNA [189], DNA [190] and enzymes [78]. Compared to the encapsulation of small biomolecules (e.g., small drugs), the adsorption of large biomolecules is more demanding. This issue to the fact that the stabilisation of the latter into the MOF pores through electrostatic interactions, which is commonly used in the case of small guest molecules, is not suitable due to their larger size. Furthermore, the maintenance of the structural and functional properties of molecules with secondary and tertiary conformations can be more challenging [191]. In the following parts the approaches for the integration of large biomolecules into MOF pores are described along with representative examples of HPMOFs categorised according to the type of biomolecule they encapsulate.

### 3.1. Strategies for the Encapsulation of Large Biomolecules by Highly Porous MOFs

The incorporation of large biomolecules with MOFs has been achieved through various methods. These include covalent attachment [192], co-precipitation [193], surface attachment [194] and encapsulation [195]. Co-precipitation involves the synthesis of the MOF in the presence of the target biomolecule, whereupon the frameworks form around the cargo. This method allows for the encapsulation of guests of many sizes, ranging from polypeptides to whole cells [196,197,198]. One major limitation of co-precipitation is the requirement that the MOF is synthesised under mild conditions which are tolerable to the biomolecules [199,200,201]. As the conventional method for MOF synthesis is solvothermal, requiring organic solvents, high temperatures and pressures there are severe restrictions on the MOFs which are compatible with this method [202]. Thus far, only MOFs from the **ZIF** family, which can be synthesised under aqueous conditions at room temperature, have been used [203,204,205,206,207,208,209,210].

Surface attachment is the most straightforward method of incorporating biomolecules with MOFs. The attachment is typically held together by weak interactions such as Van der Waals forces, hydrogen bonding and π-π stacking [211]. The advantages of surface attachment range from the versatility as a wide range of MOFs and biomolecules are compatible as well as the ease of attachment, generally requiring the stirring of the MOF in solution of the biomolecule [212,213,214,215]. The obvious drawback from this method is the weak MOF-biomolecule interactions resulting in leaching of the biomolecule during applications [213]. This can lead to tiresome purification procedures when attaching enzymes for catalysis [216].

Covalent attachment requires the post-synthetic reaction of the MOF with the biomolecule. This places some additional requirements upon the MOF: (1) the MOF must contain a functional group (NH_2_, OH, SH, etc.) that can be reacted with a biomolecule and vice versa [217,218], (2) the MOF and biomolecule must both be stable in the solvent required for the attachment as well as to any reagents used in the attachment, (3) the biomolecule must be sufficiently small to enter the pores of the MOF and react with functional groups within the MOF crystal in order to achieve maximum conversion. [219] This method has obvious advantages over surface attachment; with the formation of strong covalent bonds, there is less of a risk of leakage in addition to the well-defined positioning of the biomolecule within the MOF. Disadvantages include poor stability of the MOF and/or biomolecule under the reaction conditions, incomplete conversion with the bulk of the biomolecules being attached at or near the surface of the MOF crystal [220], the potentiality for deactivation of the biomolecule either due to the reaction and/or conformational changes in the biomolecule due to attachment [221] and the tiresome purification required to remove unreacted biomolecules, reagents, as well as possible side products. For some applications, such as the delivery of biomolecular therapeutic agents, having strong bonds between the MOF and biomolecule is undesirable as limited release can be achieved [23,222].

Biomolecules have also been used as ligands in the synthesis of MOFs, ranging from amino acids [223,224], peptides [187,188], oligosaccharides [225,226,227], nucleobases [51,123,172] as well as secondary metabolites [222]. There is a significant synthetic challenge in the synthesis due the requirement for strong metal coordinating groups in the biomolecules as well as their high flexibility making it challenging to construct rigid, crystalline and permanently porous materials. Nonetheless several MOFs have been reported with biomolecule linkers with applications ranging from catalysis [228,229], enantioselective separation [187], water purification, drug delivery and as well as acting as therapeutic agents by themselves [222]. It is worth noting that the flexibility of biomolecule linkers can also be advantageous in the design of piezoelectric MOFs [230]. One highly attractive feature is the biocompatibility and biodegradability of these MOFs (depending on metal choice) making them excellent candidates for drug delivery. Notable examples with high surface areas include **CD-MOFs** (cyclodextrin based), **medi-MOF-1** (curcumin based) (Figure 8) and the **bio-MOF** family (nucleotide based).

### 3.2. Incorporation of Enzymes into MOFs

Highly porous MOFs provide a method of entrapping enzymes, thus protecting them, whilst allowing for their use in catalysis [231]. MOFs are excellent candidates for enzyme entrapment as they allow for both hydrophobic and hydrophilic interactions, thus eliminating leaching and increasing reusability [232,233]. Thus far, many MOFs have been used to encapsulate enzymes, ranging from small enzymes such as microperoxidase-11 to the much larger Cyt C [234,235]. The mesoporous MOFs **PCN-332(M)** and **PCN-333(M)** (M = Al(III), Fe(III), Sc(III), V(III), In(III)) have been used as a single enzyme trap to encapsulate Cyt C, horseradish peroxidase (HRP) and microperoxidase-11 (MP-11). Both frameworks are isoreticular to **MIL-101(M),** being composed of aryl tricarboxylate linkers and trivalent metal ion clusters. **PCN-332(M)** is built upon the tritopic benzo-tris-thiophene carboxylate (TTPC) linker, whilst **PCN-333(M)** is built on the elongated tricarboxylate 4,4′,4″-s-triazine-2,4,6-triyl-tri-benzoate (TATB) linker (Figure 9). Both frameworks feature repeating supertetrahedral units of 11 Å internal diameter, which then generate two types of mesoporous cavities. 20 supertetrahedral units construct a 34 Å dodecahedral cage via their shared vertices **PCN-333**. A second hexacaidecahedral is constructed by 24 supertetrahedral units with 55 Å diameter. This cage is accessible via pentagonal windows and by hexagonal windows of 30 Å diameter, providing an excellent opportunity for the encapsulation of large proteins. The structure of **PCN-332** is similar with smaller windows and cavities owing to the smaller ligand [78].

**PCN-333(Al)** displayed record loadings for all three enzymes with loadings close to 1 g enzyme/g MOF. HRP is sufficiently large to only be located within the largest cavity, with only a single enzyme located in each cavity, representing an example of a single enzyme encapsulation (SEE). Cyt c is located within both the medium and large cavities, whereas MP-11 is sufficiently small to allow for multiple enzymes to be encapsulated in a single cavity. The catalytic activity of the enzymes was then studied to assess whether **PCN-333** was able to prevent leaching whilst allowing for high enzyme activity [78].

The oxidation of o-phenylenediamine by HRP@**PCN-333** and 2-2′-azino-bis(3-ethylbenzthiazoline-6-sulfonic acid) by Cyt C@**PCN-333** and MP-11@**PCN-333** were studied. The enzymes which underwent SEE (HRP and Cyt C) displayed increased activity relative to the free enzyme in water. This is due to the absence of self-aggregation and denaturation in the encapsulated enzymes. In the case of MP-11, where multiple enzymes may occupy the same cavity, decreased activity was observed, which is attributed to the aggregation of MP-11 within the pores. It is worth noting that encapsulated HRP showed much higher activity in organic solvents than the free enzymes allowing for the use of enzymes with non-water-soluble substrates. Immobilised enzymes also displayed excellent recyclability with almost no loss of activity after multiple cycles with negligible leeching [78].

Taking the encapsulation of enzymes further, Li et al. have reported a cell-free enzymatic system encapsulated by a MOF, using the enzymes lactate dehydrogenases (LDH) and diaphorase, as well as the co-enzyme nicotine-amide adenine dinucleotide (NAD^+^ and NADH) and substrate L-pyruvate (Figure 10). A series of isoreticular **csq**-net MOFs **NU1001–1007** were used for the encapsulation. The MOFs feature a hierarchical pore structure of large hexagonal pores and small triangular channels interconnected via triangular windows (Figure 10). **NU-1001–1007** were constructed from Zr_6_(μ_3_-O)_4_(m_3_-OH)_4_(OH)_4_(H_2_O)_4_ nodes and tetratopic pyrene-based carboxylate ligands (Figure 2). Specifically, the ligand arms are functionalised with phenyl (**NU-1001**), naphthyl (**NU-1002**), fluorene (**NU-1003**), stilbene (**NU-1004**), diphenylacetylene (**NU-1005**) and terphenylene (**NU-1007**) based linkers. The frameworks show progressive increase in pore and window size from **NU-1000–NU-1007** (Figure 10), highlighting the usefulness of isoreticular chemistry [101].

LDH, which has molecular dimensions of 4.4 × 4.4 × 5.6 nm, was successfully encapsulated within the large hexagonal pores of **NU-1003–NU-1007** leaving the triangular channels largely open for substrate diffusion. **NU1000–1002** contained hexagonal pores, which were too small for the encapsulation of LDH. Encapsulated enzyme activity was highest in MOFs with larger pores due to faster diffusion of co-enzyme and substrate. However, the initial rate of activity is much lower in encapsulated LDH than free LDH. This is due to requirement for the co-enzyme and substrate to diffuse into the crystals core to encounter the enzyme, the use of smaller MOF particles was sufficient to increase the rate to comparable to free LDH [101].

Upon the above observation, a cell free enzymatic system was set up, where immobilised LDH would reduce L-lactate to pyruvate with the concomitant oxidation of NADH to NAD^+^. Diaphorase then reduces NAD^+^ back to NADH. The activity of immobilised LDH increased with increasing pore size. The triangular window sizes of **NU-1007,6,5** is larger than that of NAD^+^ whilst those of **NU-1002,3,4** are smaller, thus the accessibility of immobilised LDH for NAD^+^ is significantly lower in **NU-1002,3,4** leading to lower activity. Interestingly, the rate of reaction of immobilised LDH in **NU-1005,6,7** is higher than that of free LDH. This may be due to a channelling effect whereby NAD^+^ produced within the framework readily comes in contact with diaphorase located on the surface. This is in contrast with free enzyme and diaphorase where they are randomly distributed within solution [101].

Another application of encapsulated enzymes within MOFs, is the detoxification of nerve agents as demonstrated by Li et al. The enzyme organophorphorous acid anhydrolase (OPAA) has been encapsulated within the previously mention Zr MOF **NU-1003**. OPAA catalyses the hydrolysis of highly toxic nerve agents to non-toxic products. In this case, the nerve agent simulant diisopropyl fluorophosphate (DFP) and nerve agent Soman were used as the substrates. The enzyme was loaded by simple post synthetic stirring method, where a loading of 0.12–0.2 mg/mg was observed. The size of the MOF crystals was again shown to impact the loading with smaller crystals resulting in larger loadings, in line with a diffusion-controlled process. The enzyme was primarily encapsulated within the large hexagonal pores, leaving the triangular channels open for substrate diffusion. In agreement with previous studies, large crystals (10,000—1000 nm) showed lower initial hydrolysis rates of DFP than that the free enzyme in bis-trip-propane buffer (BTP) at 7.2 pH. However, 300 nm crystals of OPAA@**NU-1003** reached 100% conversion in 2 min, a rate faster than that of free OPAA. Similar results were observed for the highly toxic nerve agent Soman. Interestingly 300 nm crystals of OPAA@**NU-1003** showed an initial reaction rate of over 3 times that of free OPAA. There are a few possible explanations for this observation, this first being a synergistic affect between the framework and encapsulated enzyme. Other explanations include a channelling effect as observed for the cell free enzymatic systems previously described and a decrease in enzyme deactivating processes such as aggregation or denaturation [21].

It Is worth noting that the usage of MOFs with flexible linkers or indeed flexible biomolecules, allows for the encapsulation of biomolecules larger than the MOF apertures. This is typically achieved by conformational changes of either component which allow the biomolecule to pass. One notable example as demonstrated by Ma et al. is the encapsulation of Cytochrome C in the mesoporous MOF **Tb-mesoMOF**. Cytochrome C is a relatively small protein, consisting of a single 104 amino acid chain and a heme group. Nonetheless it has relatively large molecular dimensions of 2.6 nm × 3.2 nm × 3.3 nm, significantly larger than those of **Tb-mesoMOF** apertures (3.2 and 4.7 nm cages accessible by 1.3 and 1.7 nm windows, respectively). Cyt C first interacts with the MOF crystal surface undergoing partial unfolding, allowing it to pass through the MOF apertures. Cyc C eventually migrates to the interior MOF cavities, where hydrophobic interactions stabilise the protein. Interestingly Cyt C adopts a conformation within **Tb-mesoMOF** that is distinct from both the original conformation and that of the denatured protein [236].

### 3.3. Incorporation of Proteins into MOFs

Numerous proteins of varying size have been incorporated into MOFs for a range of purposes, including delivery as therapeutic agents, [237] as components of hybrid MOF materials, as well as the use of MOFs as nucleation points for protein crystal growth [238]. The delivery of proteins as therapeutic agents is vital in the treatment of numerous diseases, including both type 1 and type 2 diabetes mellitus. Diabetes mellitus is one the most prevalent medical conditions in the modern world, with the main treatment requiring regular subcutaneous injection of the protein insulin [239]. As insulin is readily denatured in the stomach by acidic conditions and digested by the peptidase pepsin, it must be administer via injection. This prevents the more convenient oral delivery of insulin; thus, the development of carriers capable of resisting and protecting insulin from the conditions of the stomach is of wide interest. **NU-1000** has been shown to encapsulate insulin in significantly larger quantities than conventional carriers (40 wt%). As is well-established, Zr MOFs have excellent water and pH stability. Insulin@**NU-1000** was treated with simulated stomach fluid (pH 1 and pepsin) and showed excellent stability with only 10% release of insulin within 1 h. Subsequent exposure to PBS solution resulted in the decomposition of the framework (due to high phosphate—Zr affinity) and insulin release of 99%. The activity of encapsulated insulin versus free insulin was studied after exposure to simulated stomach acid and PBS solution and compared to untreated insulin. Remarkably, the encapsulated insulin displays similar activity to the untreated (99%) indicating that encapsulation prevents both denaturation and digestion, with the release only occurring after exposure to PBS [240].

Another application of MOFs with the regard to proteins is in the use as nucleating agents for protein crystallisation. Protein crystallisation is of vital importance to both industry and research fields, being the only method for determining the exact 3-D conformation of a protein. The crystallisation of proteins is very challenging, requiring high purity proteins and a balance between many factors such as pH, solubility, temperature, etc. [241]. Leite et al. reported that **Tb-mesoMOF** can act as a nucleating agent for the protein lysozyme (27.2 × 32.2 × 46.5 Å). **Tb-mesoMOF** was shown to encapsulate lysozyme with a loading of 3.3 μmol/g MOF. It is worth noting that the loading of two larger proteins trypsin (38 × 38 × 38 Å) and albumin (65 × 55 × 5 Å) were both attempted, but were not incorporated as the MOF pore size was inadequate. Subsequent exposure of the MOF crystals to another solution of lysozyme resulted of growth of lysozyme crystals on the MOF surface. Interestingly there was no clear interface between the MOF and lysozyme crystal. Similarly, whilst albumin was not encapsulated, albumin crystals grew on the MOF surface [22].

While most methods mentioned thus far include the encapsulation of proteins or their growth, another far less researched area is the incorporation of proteins as a component of a MOF. Although there are significant synthetic challenges (stability, flexibility of protein, multitude of metal binding sites), there have been some protein-MOFs reported with applications in sensing and catalysis [101,242]. The first reported protein-MOF by Sontz et al. is a **Zn-ferritin-bdh** (where bdh = benzene-1,4-dihydroxamic acid) MOF. Complexation of ZnCl_2_ with a mutated ferritin that includes additional histidine residues for metal-binding, led to the formation of non-polymeric fcc centred lattice, where Zn^2+^ ions are coordinated by three histidine residues and one water molecule from outside the proteins core (Figure 11). Eight Zn^2+^ ions are coordinated per ferritin molecule. This can be thus considered an SBU with very large diameter of 12 nm; the latter was subsequently reacted with a range of ditopic linkers to generate MOFs.

Interestingly commonly used bdcH_2_ did not yield a MOF but the use of bdhH_2_ resulted in the formation of a **bcc** MOF. Whilst the linker size is quite small, due to the hollow nature of the ferritin-based SBU, the framework displays a solvent content of 67%. Following this, ferritin-MOFs have been reported with the use of Ni^2+^, Co^2+^ and linkers 1,3-benzenedihydroxamic acid and 2,5-furandihydroxamic acid. These frameworks were shown to have interesting thermomechanical properties, with the Ni-ferritin-furandihydroxamate MOF undergoing an abrupt reversible transition to free ferritin and linker [243].

### 3.4. Incorporation of Nucleotides into MOFs

Whilst numerous large drugs have been successfully encapsulated in a range of microporous to mesoporous materials, highly porous MOFs allow for the delivery of therapeutics not accessible to other carriers. One such example is the encapsulation of small interfering ribonucleotides (siRNA’s). siRNA’s are double stranded RNA fragments of approximately 20 nucleotides, which can silence the expression of the complementary mRNA sequence. As siRNA only silence the expression of the complementary sequence, the use of synthetic siRNA allows for highly targeted therapy. One drawback of siRNA is the lack of stability, they are readily metabolised and destroyed by cells, thus they need to be protected from RNAase which facilitates their breakdown. In addition, as they are large molecules (7.5 nm × 2 nm), they cannot enter cells through simple diffusion through the cell membrane [189].

Fairen-Jimenez et al. have successfully encapsulated an siRNA in the mesoporous Zr MOF **NU-1000**. **NU-1000** is constructed from Zr_6_(μ_3_-O)_4_(μ_3_-OH)_4_(H_2_O)_4_(OH)_4_ SBUs and a 1,3,6,8-(p-benzoate)pyrene (H_4_TBAPy) linkers. **NU-1000** has 3 nm diameter hexagonal channels, which are larger than the smallest axis of siRNA’s, and thus capable of loading them [189,244]. Computational studies indicated that siRNA could pack within the free volume of **NU-1000** pores (Figure 12). A loading of 150 pmol/mg was achieved.

Studies were carried out to determine the degree of protection **NU-1000** provided for the siRNA. To this end, naked siRNA, siRNA@**NU-1000** and **NU-1000** were all exposed to an RNA degrading enzyme. It was shown that the bulk of siRNA was loaded within the MOF pores and thus not destroyed by the enzyme as it is too large to enter the MOF pores. Some siRNA was found to have been degraded but this is attributed to siRNA on the surface of the framework. Whilst encapsulation within **NU-1000** protected the siRNA from enzymatic degradation, it was found that the siRNA@**NU-1000** complex did not have consistent gene knockdown effects. This was attributed to the endosomal entrapment and thus experiments were carried out whereby siRNA@**NU-1000** was complexed with various factors known to breakdown endosomes. Upon complexation to these factors’ siRNA@**NU-1000** was found to have a consistent gene knockdown effect [189].

In a similar study, short single-stranded DNA (ssDNA) (less than 100 nucleotides) was successfully encapsulated in **Ni-IRMOF-74-II** to **IV** and shown to transfect immune cells with greater efficiency than currently commercially available agents. As previous mention, this family of MOFs feature 1-D hexagonal pore channels ranging from 1.5 nm for **Ni-MOF-74** to 4.2 nm for **Ni-IRMOF-74-V** (Figure 3). Maximum loadings of 6.9 wt% were observed in the cases of **Ni-IRMOF-74-III** and **IV**, the loadings are primarily driven by Van der Waal interactions between the ssDNA and linker in the pore wall. The protective effects of encapsulation were demonstrated upon exposure of encapsulated ssDNA to nuclease, with survival rates for the ssDNA over 95% in all **Ni-IRMOF-74** after 24 h exposure time, which is in stark contrast to microporous MOFs **UiO-66,67** and **MIL-101**. SsDNA-MOF interactions increased with increasing linker length showing the stronger Van der Waal interactions with the larger linkers. This is also observed in the release of ssDNA with significantly greater release in **Ni-IRMOF-74-II** than **III,IV** and **V**. As a result of the greater release, **Ni-IRMOF-74-II,III** both showed excellent intracellular gene transfection in five cell lines (two primary mouse immune cells (CD4+T cells and B cells) and two macrophages (RAW264.7 from mouse and THP-1 from human) as well as the human breast cancer cell line, MCF-7, where gene silencing was observed. These results indicate that mild DNA-MOF interactions are the most favourable, as they are strong enough to encapsulate the DNA without leakage but allowing for efficient release under the required conditions [23].

As MOF nanoparticles display poor colloidal stability [245], MOF composites are typically required for therapeutic [190] and imaging applications [246,247]. DNA functionalised spherical nanoparticles are of particular interest due to their increased colloidal stability as well as increased cellular uptake [248]. MOF-DNA nanoparticle composites have also been reported, with applications in cell targeting, imaging and intracellular protein delivery. The use of MOF-DNA composites is very desirable due to the ability to select specific DNA sequences, which in turn will have the specific interactions required for the desired application. Typically, MOF-DNA composites are stabilised by electrostatic or van der Waal interactions or bioconjugation with functional groups on the MOF linker on the surface of the MOF. This method obviously restricts the number of MOFs which are suitable for composite synthesis. Wang et al. have demonstrated a general method for the synthesis MOF-DNA nanoparticle composites via surface modification of the MOF SBUs with terminal phosphate modified oligonucleotides, followed by incubation with the complementary sequence [249]. The DNA coverage was found to correlate with the density of SBUs present on the MOF surface, the metal coordination number and the M-phosphate bond strength. The method is versatile with a range of compatible MOFs, (**UiO-66** family, **PCN-222** to **PCN-224**, **MIL-101(M)**).

MOF-DNA nanoparticles display increased cellular uptake relative to their parent MOFs [237,249]. Oligonucleotide functionalised **NU-1000** and **PCN-222** nanoparticles have been used as intercellular delivery vehicles for insulin [237]. The insulin–MOF nanoparticles were DNA functionalised by the method described above. The resulting spherical nucleic acid nanoparticles display a 10-fold increase in cellular uptake relative to pure insulin. The MOF-DNA nanoparticles then release their cargo due to the increased intracellular phosphate concentration. In a similar study Wang et al. functionalised **nano-UiO-66-NH_2_** with specific DNA sequences, which resulted in targeted release of the DNA in organelles with high phosphate concentrations [190]. This method can allow for targeted and controlled release in combination with increased cellular uptake.

## 4. Conclusions

In this review, the synthetic approaches towards the achievement of HPMOFs, and their current progress within the field of therapeutic biomolecules, have been presented. The first member of this family, **MOF-177,** with a surface area of 4750 m^2^ g^−1^, was reported in 2003 by Yaghi et al. Since then, this family has been ever-developing, reaching the current most highly porous MOF, **NU-1501,** which exhibits a surface area of 7310 m^2^ g^−1^. There are now over 45 HPMOFs known that exhibit a surface area higher than 4000 m^2^ g^−1^, which are listed in Table 1. The increase in porosity of the framework allows for an increase in guest molecule accessibility, and therefore HPMOFs are promising materials for large molecule incorporation and expanding these materials’ application in many fields of biosciences including delivery of biopharmaceuticals, gene targeting and the crystallisation of proteins.

The strategies employed to incorporate biomacromolecules with HPMOFs is the main objective of this review because of the increasing demand within the biomedical area for materials capable of delivering these therapeutic biomolecules with advanced efficiency and efficacy. Several approaches have been developed for the incorporation of large bulky biomolecules, including covalent attachment, surface attachment and encapsulation. Although studies of the co-precipitation strategy have been conducted, this approach was only briefly mentioned in this review because the **ZIF** family of MOFs are the only MOFs to have incorporated biomolecules via this method and are not highly porous. However, this approach remains as a potential for biomolecule incorporation by HPMOFs.

The hydrophobic and hydrophilic nature of MOFs permits their excellent candidacy for enzyme entrapment, leaching prevention and improved reusability. The research of HPMOFs and enzyme incorporation entails many interesting aspects including single enzyme encapsulation by **PCN-333(Al)**, the conformationally changes an enzyme undergoes to enter **Tb-mesoMOF**, the catalysis mechanisms by OPAA**@NU-1003** and the micro-mesoporous hierarchical MOF **NU-1005,6,7** capable of multienzyme encapsulation. In regard to proteins, HPMOFs have been discovered to act as a nucleating agent for the promotion of protein crystallisation and to be amalgamated with protein molecules creating protein-MOF composites. Protein crystallisation was induced by firstly encapsulating the enzyme, lysozyme, by **Tb-mesoMOF** to form protein crystals. The production of a protein-MOFs poses many challenges because of protein stability and flexibility. However, the HPMOF, **Zn-ferritin-bdh**, represents the beginning of the exciting potential of protein-MOF materials. HPMOF research with nucleotide incorporation has led to materials capable of stabilising DNA in solution and delivering nucleotides with therapeutic potentials into cells. siRNA@**NU-1000** establishes the potential of MOFs for siRNA stabilisation, protection from enzyme degradation and gene knockdown effect. **Ni-IRMOF-74-II,III** demonstrates the potential of HPMOFs for gene silencing. Additionally, **NU-1000** and **PCN-22**, functionalised with oligonucleotides illustrate that HPMOFs can act as delivery vehicles for insulin with impressive outcomes such as a 10-fold increase in cellular uptake in comparison to pure insulin. The utility of all these incorporations will advance structure-based drug design and expand the biomedical fields of biocatalysis, protein therapeutics and cancer-causing gene knockdown research.

Although this review focuses on the biomedical area regarding HPMOFs, it is important to note that these HPMOFs also have gas sorption capabilities [246,247]. HPMOFs find applications in gas separation for purification, gas storage for fuel, gas delivery for biomedical applications and gas capture for environmental purposes. MOFs are promising sorbents for gases because of the high surface areas, pore volume, tuneability and crystalline nature that these materials exhibit. Gas capture studies with MOFs include H_2_ [248], CH_4_ [249,250,251], NO*_x_* [252], CO_2_ [253] and CO [254]. It is noteworthy to mention that microporous MOFs offer advantages compared to HPMOFs when it comes to gas uptake, owing to their utility at low pressures and smaller pore sizes that allows for enhanced interactions with the framework instead of interactions with other gas molecules that are bound to the framework. Other, newly emerging applications of HPMOFs include CO_2_ fixation [250,251,252]), electrocatalytic methanol oxidation reaction [253], and dyes adsorption [254,255], with a prominent representative example of the latter being the anionic, pyrene-based MOF [BMI]_2_[Mg_3_(TBAPy)_2_(H_2_O)_4_]·2dioxane, which exhibits improved light-harvesting and photoelectric conversion efficiency by the encapsulation of D−π−A cation dyes [254].

Whilst HPMOFs are highly promising candidates for the delivery of biotherapeutics significant work is required to establish MOF toxicity and biodistribution. To investigate MOF toxicity many factors must be considered. The toxicity of the metal and linker play a crucial role, however, particle size effects, colloidal stability, aggregation, biodistribution, accumulation and metabolism greatly affect toxicity [256,257]. To date, in vivo and in vitro studies have only been carried out on a small number of MOFs, mainly those containing small carboxylate linkers. Nonetheless, some toxicity trends are observed. In general, MOFs with poor aqueous stability and exogenous linkers display the highest toxicity [258]. This toxicity is rationalised by the rapid release of metal ions in sufficiently high concentrations to exhibit toxic effects. Thus, Zr, Ca and Fe MOFs display surprisingly low toxicity relative to other metals (Zn, Cu, Mn). In the case of HPMOFs, toxicity is most likely to originate from the larger hydrophobic linkers. Studies have shown that hydrophilic MOF linkers are rapidly excreted, displaying lower toxicities than hydrophobic linkers [259]. For the future prospective of the biomedical application of HPMOFs, it is essential that pharmacological studies are performed to allow for translation into clinical applications.

The pathway towards HPMOFs has involved challenges such as interpenetration and poor stability. However, recently there has been progress in the approaches towards overcoming these challenges, resulting in the generation of HPMOFs. This review endeavours that the targeted synthesis methods outlined will serve as a useful guide for the generation of breakthrough HPMOFs. Isoreticular expansion (**bio-MOF-100**), topological control (**PCN-61**, **PCN-66**, **PCN-68** and **PCN-610**), MOPs as SBBs (**PCN-82**) and controlled defect formation (**PCN-160**) are model strategies for HPMOF attainment. It must be mentioned that there are other synthetic strategies such as the pillar MOF approach and the solvent assisted ligand exchange method (SALE), that represent as potential platforms for HPMOFs achievement. The pillared MOF approach involves the employment of one ligand during synthesis to access a 2D framework, while an additional ligand is added to bridge the 2D layers, resulting in the formation of a highly porous MOF [246]. SALE is a post synthetic method that involves the selection of a geometrically similar but longer ligand to the shorter daughter ligand of the MOF in question, followed by a single-crystal-to-single-crystal ligand exchange and the formation of a new MOF [46,173]. All the strategies highlighted in this review have resulted in the generation of a database consisting of over 45 HPMOFs.

In conclusion, HPMOFs have generated exciting discoveries within many fields of science. The ever-growing database of HPMOFs and development of efficient targeted synthetic approaches towards HPMOFs can serve as inspiration for discovering novel HPMOFs with potential for even higher porosities than what has been achieved and/or reaching the theoretical upper limit of 14,600 m^2^ g^−1^ [7]. In addition, the environmental application of HPMOFs might prove an important area for future research in regard to dye uptake and metal extraction from aqueous systems. The future direction of HPMOFs and their applications in gene therapy, therapeutic protein transport and biocatalysis is full of exciting and fruitful possibilities. With the aid of HPMOFs, the notoriously difficult stabilisation, crystallisation and delivery of therapeutic biomolecules will be conquered, thus enabling the development of breakthrough discoveries.

## Data Availability

Not applicable.

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
