# Peer review of "Synthesis and Biomedical Applications of Highly Porous Metal–Organic Frameworks"

_molecules, 2022, doi:10.3390/molecules27196585_

Round 1

Reviewer 1 Report

In the manuscript by Papatriantafyllopoulou et al., the authors have made an effort to introduce the chemical aspects of the biomedical applications of HPMOFs. After carefully reading the manuscript, I believe it is suitable for publication in molecules after carefully considering the comments below:

-The authors did not mention that the low toxicity of HPMOFs throughout the manuscript. However, different organic linkers may also display different toxicity or trigger different immune responses. In this regard, it is not clear to the reader how the toxicity of T HPMOFs is assessed and what linkers should be used to ensure good biocompatibility.

-In the conclusions, I suggested the authors also state that the surface-modified HPMOFs. I believe they mean higher porosity?

-There are serious grammatical errors in this manuscript. For example, the following expression in the 3rd line from the bottom on page 28 - In the concluding section, it is necessary to describe in more detail the prospects for the biomedical application of HPMOFs.

-In the introduction, the author could compare and cite the refs on HPMOFs, such as Inorg. Chem., 2022, 61, 9328-9338. Inorg. Chem. 2021, 60, 18593-18597; Cryst. Growth Des. 2022, 22, 4018-4024; Inorg. Chem. 2022, 61, 13234-13238; Chem. Commun., 2020, 56, 8758−8761; Angew. Chem. Int. Ed., 2019, 58, 12185-12189. Chem. Commun., 2020, 56, 2395−2398.

- The author should list the different strategies by loading and compare the different feature.

Author Response

please see pdf attached

Reviewer 2 Report

Comments:

This paper presents a review on the synthesis of highly porous metal-organic frameworks, framework topology and biomedical applications. Attempts were made to classify some of the main synthetic methods of HPMOFs. Describes HPMOFs with capability to encapsulate large biomolecules, along with description of the encapsulation strategies. Finally, some conclusions in this area of chemistry are presented, and predicts an exciting future for the application of HPMOFs. Although the authors provide a very nice review of HPMOFs,there are some details issues. Authors should carefully revise these questions to meet the standards of molecules journal.

Some problems/suggestions are listed below:

1.   Whether the keywords are a bit much?

2.   As a review, authors should cite the most recent articles from the last few years

3.   Authors should indicate the cited articles about the drugs and personal care products of line 46

4.   Can the author explain the meaning of the SBUs that appear in line 151?. Please make sure the same problem does not occur again。

5.     What is the benefit of the MOF materials compared to other materials? It should compare and discuss it, especially porous polymers. Advanced functional materials, 2020, 30(2): 1902634. Biomater. Sci., 2022, DOI: 10.1039/D2BM00719C.  J Control Release. 2022 Aug 6:S0168-3659(22)00485-0. doi: 10.1016/j.jconrel.2022.08.005.

Author Response

please see pdf attached

Round 2

Reviewer 1 Report

the authors have addressed all the comments.